# MathDSL: A Domain-Specific Language for Concise Mathematical Solutions Via Program Synthesis

**Sagnik Anupam**
MIT
sanupam@mit.edu

**Maddy Bowers**
MIT
mlbowers@mit.edu

**Omar Costilla-Reyes**
MIT
costilla@mit.edu

**Armando Solar-Lezama**
MIT
asolar@csail.mit.edu

## Abstract

We present MathDSL, a Domain-Specific Language (DSL) for mathematical equation solving, which, when deployed in program synthesis models, outperforms state-of-the-art reinforcement-learning-based methods. We also introduce a quantitative metric for measuring the conciseness of a mathematical solution and demonstrate the improvement in the quality of generated solutions compared to other methods. Our system demonstrates that a program synthesis system (DreamCoder) using MathDSL can generate programs that solve linear equations with greater accuracy and conciseness than using reinforcement learning systems. Additionally, we demonstrate that if we use the action spaces of previous reinforcement learning systems as DSLs, MathDSL outperforms the action-space-DSLs. We use DreamCoder to store equation-solving strategies as learned abstractions in its program library and demonstrate that by using MathDSL, these can be converted into human-interpretable solution strategies that could have applications in mathematical education.

## 1 Introduction

Building machine learning models that can replicate human reasoning abilities in symbolic domains, such as algebra or arithmetic, is a challenging problem that researchers continue to face today [7]. Even large models that exhibit state-of-the-art performance on language modelling datasets, like GPT-4, exhibit much poorer performance on mathematical exams involving reasoning tasks [6]. Improving these systems to perform well on reasoning tasks often requires extensive usage of techniques such as chain-of-thought reasoning and training on large datasets of mathematical data, which is expensive in both time and cost.

Improving machine learning models' mathematical reasoning ability may yield significant interpretability that can provide educational benefits, as studies have shown that automated tutor systems can yield similar or greater educational gains than human tutors [11]. Additionally, such improvements, when extended to more complex mathematical domains, can aid in developing software for researchers by helping describe the behavior of previously unknown functions [4]. Thus, an important research goal is to develop machine-learning systems that write step-by-step solutions to mathematical problems while being as efficient as possible regarding training data and compute.

For simple algebraic domains like linear equations, many step-by-step solvers rely on manually written heuristics. However, those have been shown to have surprising blind spots that fail to account for solving rare equation cases [7]. In recent years, attempts to help models learn neurosymbolic reasoning in mathematical domains have led to the development of specific reinforcement learning

---

Code available at: https://github.com/sagnikanupam/mathdsl

Preprint. Under review.

techniques, like the Contrastive Policy Learning (ConPoLe) algorithm [7]. When presented with an equation, ConPoLe uses reinforcement-learning techniques to query the search space and find the most promising next step until it synthesizes a complete step-by-step solution.

However, in the linear equations domain, the solutions generated using ConPoLe often tend to be elaborate and unwieldy [5], and often contain unnecessary steps that may confuse program users, as demonstrated by the ConPoLe solution in Figure 1. Recent research has demonstrated that humans find solutions simplified using skill-based, higher-level abstractions very useful [9]. One approach that uses this idea is a theorem-proving language, Peano [8], which changes ConPoLe's action space to a finite set of valid axioms and discovers tactics by analyzing a batch of ConPoLe-generated solutions. However, solution discovery in Peano is limited to a few types of equations due to the limitations of the tactic language, and in practice, the discovered tactics are quite simple [8]. Another approach, Lemma [5], leverages the idea of abstraction learning by examining several ConPoLe solutions generated on a training dataset (using its original action space) and building abstractions that ConPoLe can use to solve equations on a test dataset. However, Lemma requires training on a large dataset of previously generated ConPoLe solutions to build high-quality abstractions. Hence, it is not able to leverage the power of the abstractions the first time it solves tasks in a new domain.

Our approach, which pairs a Domain-Specific Language (DSL) with the DreamCoder program synthesis system [2], overcomes this hurdle by generating powerful abstractions during the training process that help develop more concise solutions and improve accuracy without requiring a large dataset of solutions. We demonstrate that existing action spaces are not effective DSLs for DreamCoder. Instead, we develop an expressive DSL, which we name MathDSL, and use DreamCoder with the Stitch library learning algorithm [1] to enumerate potential solution programs and discover abstractions. DreamCoder can then use these MathDSL abstractions to solve new, heldout equations, and build a hierarchy of high-level abstractions in future training iterations.

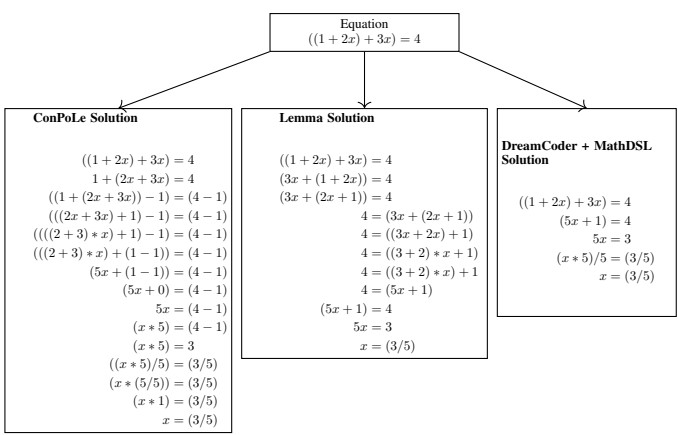

Figure 1: Comparison of ConPoLe solutions with more concise DreamCoder+Stitch+MathDSL and Lemma solutions. The ConPoLe solution contains unnatural subroutines, while DreamCoder+MathDSL and Lemma offer more human-like strategies.

## 2 Methods

Previous research on developing machine-learning models for mathematical equation solving has shown that neural models perform poorly on arithmetic tasks unless task-specific components are used [13]. To improve the performance of neural models, recent approaches have examined if it is possible to use these models in sequence-to-sequence contexts by first converting mathematical expressions into Abstract Syntax Trees (ASTs), and then into prefix sequences derived from these trees [4]. Extending this idea, MathDSL modifies the problem domain from algebra to *algebraic manipulation* and uses primitives that describe operations performed on the prefix form of equations.

Our approach involves searching the space of all possible programs in MathDSL to find the correct program that converts the input from the prefix form of the equation to the solution state string. Thus, the entire solution to the equation is synthesized at once, unlike previous reinforcement learning systems, which attempt to find the next step in the solution by choosing from the existing action space. For searching the space of MathDSL programs, we use DreamCoder [2], a library-learning framework designed for use on inductive program synthesis problems [12]. We use DreamCoder with the Stitch library learning algorithm [1] (as described in [3]). A complete description of the program synthesis system and its components has been provided in Appendix A.

We provide DreamCoder with MathDSL, our DSL that encodes a set of mathematical axioms, along with some elementary equation manipulation operations, as basic programming primitives. These primitives can then be composed to form more complex operations that can fully convert a linear

equation into its corresponding solution state. MathDSL performs manipulations on equations in prefix form and comprises three types of primitive operations: *tree operations*, *arithmetic operations*, and *index operations*. *Tree operations*, such as tree rotations and distributive operations, are defined as operations performed on the equation's tree structure that rearrange the equation tree to make future simplifications easier without introducing additional nodes. *Arithmetic operations*, like dividing or multiplying both sides of the equation by a term, introduce additional nodes in the equation tree. *Index operations* are helper operations used to help determine the indices for accessing the specific subtrees of an equation AST on which tree and arithmetic operations act. In MathDSL, arithmetic operations can only accept arguments that are subtrees of the current tree, and tree operations can only be performed on the current equation tree's subtrees. This constraint helps us avoid having to discretize the space of integers or real numbers in the DSL, which would make the search space much larger and make program synthesis prohibitively expensive. In our experiments, the index operations in MathDSL can generate integer constants from 0-110, a range much larger than the number of subtrees in any equation in our dataset. The complete list of MathDSL's primitive operations is provided in Appendix B, and an example program for solving an equation is provided in Figure 2.

## 3   Conciseness Metric

Here, we describe a metric to measure the conciseness of solutions generated by different equation-solving systems. We use an AST-based metric to ensure that large expressions were not added or transposed to different sides of the equations. We define the metric function as follows, for a solution $s$ with $n$ steps of an equation $e$:

$$f(s) = \sum_{i=1}^{n-1} max(|s_i.left - s_{i+1}.left|, |s_i.right - s_{i+1}.right|, 1)$$

Here, $s_i$ refers to the equation tree of the equation at the $i$-th step of the solution, while $x.left$ and $x.right$ refer to the **size** of the left and right subtrees of $x$, respectively, where the size is defined as the number of subtrees of a tree (including the original tree). For a given equation, a solution with a smaller value of $f(s)$ indicates a more concise solution.

As demonstrated in Figure 1, solutions that introduce large terms into the equation tree without prior simplification make the solution more difficult to interpret, especially in a pedagogical context. Hence, this metric function penalizes solutions that introduce complicated expressions in a solution step. Since this function assumes a cost of at least 1 occurring per step and does not normalize for length, it rewards shorter solutions over longer solutions. As a result, the function penalizes solutions that take many steps to simplify expressions without changing tree size (such as the ConPoLe solution in Figure 1). However, it can also reward overly-compressed solutions. A description of metric function limitations can be found in Appendix C.

For a given equation $e$, a target model $A$, and a baseline model $B$, we use the metric function to compare two solutions to the same equation by measuring the relative improvement or decline in the metric compared to the baseline model. Let $s_A$ and $s_B$ denote the solutions generated by the target model and the baseline model, respectively. Then, the relative improvement or decline in performance is measured by the score:

$$C(s_A, s_B | s_A, s_B \text{ both solve } e) = \frac{f(s_B) - f(s_A)}{f(s_B)}$$

We refer to this score as the C-score of $A$ and $B$ on $e$. We report the mean C-score over all equations solved by both the target and the baseline models. Positive mean C-scores indicate that the target model generates more concise solutions than the baseline model on average. In contrast, negative mean C-scores indicate that the target generates less concise solutions than the baseline on average. To measure conciseness in our experiments, we take ConPoLe as our baseline model and compute the C-scores of Lemma and DreamCoder (considering each DSL separately) with respect to ConPoLe.

## 4   Experiments and Results

For evaluating the performance of MathDSL in our program synthesis system and compare its performance to Lemma and ConPoLe in terms of accuracy and conciseness, the models were evaluated on a variant of the Cognitive Tutor Algebra dataset [7]. We note that it is extremely difficult to normalize the performance of DreamCoder and its reinforcement learning alternatives,

since DreamCoder has previously shown the ability to learn useful abstractions from a few carefully chosen tasks [2], while both ConPoLe and Lemma rely on generating $\approx 10^6$ equations from the equation templates during training. Instead of training Lemma and ConPoLe from scratch on our training set, which may reduce their efficacy by training them on a smaller set of equations, we simply evaluated the final trained models on the infix-notation form of our train and test datasets to observe if the models can discover the equation solutions. Our evaluation also allowed Lemma to use all the abstractions it had discovered during its original training. Additional details about the experiments are in Appendix D, and some abstractions discovered by DreamCoder are described in Appendix E.

Additionally, to demonstrate that the improved performance is due to using the MathDSL and not due to DreamCoder being an inherently more powerful system, we also run experiments with DreamCoder using ConPoLe and Lemma's action spaces as DSLs. In our first experiment, we treat each axiom listed by [7] in Appendix A as a separate primitive in a new DSL (titled ConPoLeDSL). In our second experiment, in addition to the aforementioned axioms, we use the 15 abstractions discovered by [5] as primitives in another DSL (titled LemmaDSL). The performance of the different model experiments (DreamCoder (with MathDSL, ConPoLeDSL, and LemmaDSL), ConPoLe, and Lemma) in terms of accuracy and the conciseness metrics (evaluated by comparing against ConPoLe as the baseline model) are presented in Table 1.

| Model Name | Training Set Accuracy | Testing Set Accuracy | Training Set Mean C-Score | Testing Set Mean C-Score |
|---|---|---|---|---|
| DreamCoder + Stitch + MathDSL | **0.9192** | **0.9070** | **0.5921** | **0.4859** |
| Lemma* | 0.7980 | 0.8488 | 0.5758 | 0.4752 |
| ConPoLe* | 0.8182 | 0.8372 | *0.0* | *0.0* |
| DreamCoder + Stitch + ConPoLeDSL | 0.0707 | 0.1047 | -0.7932 | -1.3611 |
| DreamCoder + Stitch + LemmaDSL | 0.3182 | 0.3488 | 0.5074 | 0.3829 |

Table 1: Accuracy and C-Scores (out of 198 randomly sampled training set problems and 86 test set problems). The * indicates that the model was not trained on this exact training set. Italicized text indicates that the ConPoLe solutions were the baseline solutions, and are expected to have a Mean C-Score of 0.

As expected, both Lemma, DreamCoder+Stitch+MathDSL, and DreamCoder+Stitch+LemmaDSL's conciseness metrics have positive values, since they generate shorter solutions than ConPoLe does. A ConPoLe target with a ConPoLe baseline will give C-scores of 0 for all equations as $s_A = s_B \implies f(s_A) = f(s_B)$. Since ConPoLeDSL and LemmaDSL have a large number of primitives performing small modifications to the equation, solution programs in those DSLs are long and difficult to discover, leading to lower overall accuracy. Thus, we can conclude that DreamCoder, when used with MathDSL, is able to generate solutions to a large number of equations in the dataset using much fewer training examples as compared to ConPoLe and Lemma. Additionally, the solutions generated by composing DreamCoder's MathDSL abstractions together tend to be much more concise in nature as evidenced by the average percentage decrease in the metric function value when compared with baseline models such as ConPoLe and Lemma. We also observe that after further post-processing of DreamCoder solutions to remove identical steps (caused by intermediate abstractions that do not change the state of the program), the mean C-Scores of MathDSL increase to 0.6237 for the training set and 0.5521 for the test set respectively. Further details on the solution generation procedure and the step de-duplication are provided in Appendix D. These results confirm that MathDSL, when combined with DreamCoder, facilitates the discovery of concise, reusable abstractions, leading to more efficient, human-interpretable and accurate equation-solving strategies.

## 5  Conclusions

In this paper, we introduced MathDSL, a domain-specific language designed for solving linear equations via program synthesis. Our results demonstrate that DreamCoder combined with MathDSL significantly outperforms existing models like ConPoLe and Lemma in terms of both accuracy and solution conciseness. Specifically, DreamCoder+MathDSL achieved a testing set accuracy of **90.70%** and produced solutions that were on average **48.59%** more concise than those generated by ConPoLe, as evidenced by the C-score metric. Furthermore, MathDSL proved to be highly efficient in terms of training data, requiring only **198** equation templates compared to the $\sim 10^6$ equations required by ConPoLe and Lemma. The system also produced human-interpretable solution strategies, unlike ConPoLe, which lacks a structured abstraction mechanism to yield human-interpretability.

Our experiments using ConPoLeDSL and LemmaDSL confirmed that MathDSL's design, rather than the underlying DreamCoder framework alone, was responsible for the improved performance. The experiments showed that neither ConPoLeDSL nor LemmaDSL enabled DreamCoder to generate concise solutions with comparable accuracy.

In conclusion, MathDSL offers a novel approach for generating interpretable and concise solutions to mathematical equations, with broader applications in educational tools and automated reasoning systems. Future work can explore extending MathDSL to more complex mathematical domains, such as calculus or discrete mathematics, and its potential applications in automated tutoring systems where human-like solution strategies are valuable.

## Acknowledgments and Disclosure of Funding

The authors would like to thank G. Poesia and Z. Li for helpful discussions, and O. Bastani for generous access to compute. S.A. was supported by the MIT Undergraduate Research Opportunities Program (UROP) and the MIT Advanced Undergraduate Research Opportunities Program (SuperUROP). This paper is a part of the Understanding the World Through Code project, supported by the National Science Foundation under Grant No. 1918839.

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

# A   Program Synthesis System Description

For performing program synthesis in our domain-specific language, we use DreamCoder [2], a wake-sleep learning algorithm, alongside Stitch [1], a library-learning algorithm. DreamCoder comprises an initial library $\mathcal{L}_0$ containing the "base primitives" defined in the domain-specific DSL, which in our case would refer to MathDSL, ConPoLeDSL, or LemmaDSL. At the end of the process, it returns a learned final library, $\mathcal{L}_f$, which contains the various subroutines (or abstractions) generated by composing the program primitives, which aided it in solving multiple tasks in the domain. By examining the new primitives DreamCoder assigns to $\mathcal{L}_f$, we can study the equation-solving procedures and algorithms it has "learned", and compare them to similar human techniques. Additionally, DreamCoder also returns a neural search model $\mathcal{Q}(\rho|t, \mathcal{L})$. When given a task $t$ and library $\mathcal{L}$, $\mathcal{Q}$ can generate several potential solution programs $\rho$ which have a high probability of solving the given task [2].

In order to construct $\mathcal{Q}$, DreamCoder first assigns a real-valued weight $\theta_{\mathcal{L}}$ to each library function $l \in \mathcal{L}$. Then, for every $l$, after normalizing weights, we have a production probability $P[l|\mathcal{L}, \theta_{\mathcal{L}}]$. Then for any program $\rho$ comprising of library functions $l$, we have prior probability of $\rho$ given by:

$$P[\rho|\mathcal{L}, \theta_{\mathcal{L}}] = \prod_{l \in \rho} P[l|\mathcal{L}, \theta_{\mathcal{L}}] \tag{1}$$

Then, a neural search model $\mathcal{Q}$ is constructed to predict the programs that can solve a given task $t$ using functions from the current library [12]. This probability is expressed as $P[\rho|t, (\mathcal{L}, \theta_{\mathcal{L}})]$ (probability of program conditioned on task, library, and library weights), so the model can be expressed in the following manner at iteration $i$ of training:

$$Q_i(\rho|t, \mathcal{L}_i) \approx P[\rho|t, (\mathcal{L}_i, \theta_{\mathcal{L}_i})] \propto P[t|\rho] P[\rho|(\mathcal{L}_i, \theta_{\mathcal{L}_i})] \tag{2}$$

Here, the probability $P[t|\rho]$ is the likelihood the task is solved by the program under consideration. Thus, programs are sampled from the prior and executed to compute the posterior probabilities of solving tasks. Then, $Q$ is trained to assign higher probabilities to programs with higher posterior probabilities [2].

DreamCoder divides this program into the following three stages: wake, sleep(abstraction), and sleep (dream). Firstly, in the wake stage, a random minibatch of tasks is sampled, and programs with higher posterior probability are found using beam search.

Secondly, in the abstraction stage, the programs discovered via beam search are held constant, and the library is rewritten to include new primitives that compress both the programs as well as the description length of the updated library. For refactoring, we use Stitch, a corpus-guided top-down synthesis algorithm which has been shown to outperform DreamCoder's library rewriting algorithm in certain domains [1]. Stitch uses a utility function for scoring abstractions given a corpus and a rewrite strategy. This utility function is calculated in Equation 17 in [1] as follows:

$$U_{\mathcal{P}, \mathcal{R}}(A) \approx -\text{cost}(A) + \sum_{p \in \mathcal{P}} \max_{e \in \text{subtrees}(p)} \text{cost}(e) - \text{cost}(\text{REWRITE}(A, e)) \tag{3}$$

In this equation, $\mathcal{P}$ is the corpus while $\mathcal{R}$ is the rewrite strategy, while $A$ is the abstraction for which utility is being computed. The summation term in the equation is summed over all programs $p$ in the corpus $\mathcal{P}$.

Thirdly, in the dream stage, $\mathcal{Q}$ is trained to assign a high probability to programs $\rho$ with higher posterior probability. This is done by having DreamCoder generate its own examples: "replays" of tasks it has already solved successfully, and "fantasies" of tasks which it creates on its own. These fantasies are created by generating programs by combining various primitives in its own current library, executing these programs on various sample inputs to generate outputs, and passing the (input, output) data points to the neural model to train it until it learns to accurately compute the probability for programs [2].

# B Description of MathDSL

MathDSL's primitive operations are listed in Table 2, alongside their type annotation (which Dream-Coder utilizes for synthesis). Table 2 also describes whether an axiom comparable to our primitive is present in ConPoLe's action space, while Table 3 describes ConPoLe axioms that were omitted from MathDSL since they can be expressed using semantically equivalent programs. In the primitive name, given an equation $e$ to solve and an integer $x$, we define $y$ as a subtree of $e$ uniquely identified by the integer $x$. Then, the complete list of all primitive operations in the DSL is as follows:

| Primitive Name | Primitive Type Annotation | Primitive Description | Equivalent Primitive Present in ConPoLe |
|---|---|---|---|
| $\text{add}(x, e)$ | tstr → tint → tstr | arithmetic operation adding $y$ to both sides of $e$. | Yes |
| $\text{sub}(x, e)$ | tstr → tint → tstr | arithmetic operation subtracting $y$ from both sides of $e$. | Yes |
| $\text{mult}(x, e)$ | tstr → tint → tstr | arithmetic operation multiplying $y$ on both sides of $e$. | Yes |
| $\text{div}(x, e)$ | tstr → tint → tstr | arithmetic operation dividing $y$ on both sides of $e$. | Yes |
| $\text{newConstGen}(a, b, c)$ | tint → tint → tint | index operation accepting numbers $a$, $b$, and $c$ and returning an integer $(a*b)+c$. | No |
| $\text{lrotate}(x, e)$ | tstr → tint → tstr | performs a left rotation on $y$ to create $y'$ and replaces $y$ with $y'$ in $e$. Also adjusts operations according to operator hierarchy and associativity rules. | No |
| $\text{rrotate}(x, e)$ | tstr → tint → tstr | performs a right rotation on $y$ to create $y'$ and replaces $y$ with $y'$ in $e$. Also adjusts operations according to operator hierarchy and associativity rules. | No |
| $\text{swap}(x, e)$ | tstr → tint → tstr | swaps the left and right children of $y$ to create $y'$ and replaces $y$ with $y'$ in $e$. | Yes |
| $\text{dist}(x, e)$ | tstr → tint → tstr | applies the distributive property $(ab + ac) = a(b + c)$ on $y$ (if applicable) to create $y'$ and replaces $y$ with $y'$ in $e$. | Yes (present within the dist ConPoLe axiom) |
| $\text{revdist}(x, e)$ | tstr → tint → tstr | reverses the distributive property $a(b + c) = ab + ac$ on $y$ (if applicable) to create $y'$ and replaces $y$ with $y'$ in $e$. | Yes (present within the dist ConPoLe axiom) |
| $\text{simplify}(x, e)$ | tstr → tint → tstr | simplifies $y$ to create $y'$ and replaces $y$ with $y'$ in $e$. Simplification involves enforcing several mathematical axioms (e.g., simplifying constants and eliminating redundant terms). | Yes |
| | | - If $A$ and $B$ are constants, simplifies $A + B, A - B, A * B, A/B$ to a single constant. | Yes |
| | | - If subtree $x$ is of form $A + 0$, $A - 0$, $A * 1$, or $A/1$, simplify subtree to $A$. | Yes |
| | | - If subtree $x$ is of form $A - A$ or $A * 0$, simplify to 0. | Yes |
| | | - If subtree $x$ is of form $A/A$, simplify to 1 (if $A \neq 0$). | Yes |
| addzero | tstr → tint → tstr | Adds zero to the right side of $y$. | Yes |
| subzero | tstr → tint → tstr | Subtracts zero from the right side of $y$. | Yes |
| multone | tstr → tint → tstr | Multiplies one on the right side of $y$. | Yes |
| divone | tstr → tint → tstr | Divides by one on the right side of $y$. | Yes |

Table 2: Description of Primitives in MathDSL

| ConPoLe Axiom Name | Equivalent Action in MathDSL |
|---|---|
| refl | We don't need to encode reflexive equality since our commutativity operator, swap, rearranges $2 = x \implies x = 2$. |
| assoc | Associativity rules are considered while performing left and right rotations, so a separate operation is unnecessary. |
| sub_comm | Left and right rotations handle reordering arguments within the same operation for all four arithmetic operations (e.g. $a+b+c = (a+c)+b$). |
| sub_sub | The model can insert 0 via swap and addzero or subzero and simplify the constants to convert from positive to negative constants and vice versa. Hence the equivalence between $+(-x)$ and $(-x)$ does not need to be explicitly encoded. |
| zero_div | While zero divided by a non-zero constant will always be 0 arithmetically, for an expression $x$, our system can instead multiply $0/x$ by $x$ and rotate to obtain $0 * (x/x)$ and then simplify to get $0 * 1 = 0$. |

Table 3: Primitives in ConPoLe's Action Space Not Reused in MathDSL

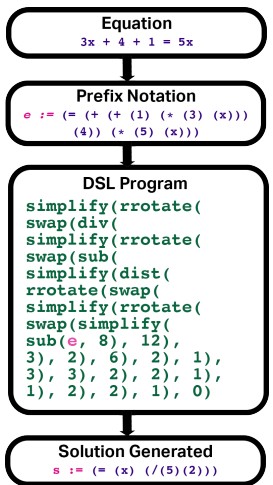

Figure 2: Example of an equation, its corresponding prefix form string, a solution program solving the equation written in MathDSL, and the solution string generated when the equation string is passed as input to the solution program. An example of a useful abstraction here would be `simplify(rrotate())`, which could then be composed with other abstractions to build more powerful abstractions in future training iterations.

## C   Limitations of Conciseness Metric

There are two main drawbacks of the conciseness metric function $f$ described in Section 3. The first drawback is that since it tends to generally reward shorter solutions, it may provide solutions with large jumps, like $Ax + B = C \implies x = (C - B)/A$, with similar scores as solutions like $Ax + B = C \implies Ax = C - B \implies x = (C - B)/A$, even though the latter is arguably a clearer and more explanatory solution.

We argue that our methods allow for the latter to be easily recoverable from the former since our abstractions are comprised of functions composed together. Hence we can decompose the functions in the abstraction as we wish, to adjust for the mathematical maturity of the student, and thus we can obtain the latter solution from the former. On the other hand, we cannot assign equal metric function values to a solution like $2x + 5 = 7 \implies 2x + 5 - 5 = 7 - 5 \implies 2x + 0 = 7 - 5 \implies 2x = 7 - 5 \implies x = 1$ and a solution like $2x + 5 = 7 \implies 2x = 7 - 5 \implies x = 1$ and say that the abstractions required to generate the latter solution are easily recoverable from the former. The statement does not hold because if the abstractions are not already present in the learned library of the model, it may lead to the former solution never being discovered by the program synthesis model in the first place. The equation could very well not be solved within the time specified since the syntax tree of the program required to generate the former solution is too deep, and thus never be found by the program synthesis model.

The second drawback is that the metric function is not as helpful for comparing solutions to two different equations, since equations have different types and different original equation tree sizes. Complex equations with larger tree sizes will require more steps to solve than simpler equations, and as a result, we observe that the scores are not directly comparable. However, we have developed the notion of C-Scores in Section 3 to compensate for this drawback, as we simply take the mean of the percentage difference in the metric function value on solved problems across the entire dataset. This approach helps measure the performance improvement of a model across a wide variety of problems, thus accounting for the difficulty in directly comparing two different solutions.

Future work could explore refining the metric to better account for pedagogical clarity and solution complexity while maintaining the efficiency of program synthesis. Approaches for overcoming these limitations may involve adding additional terms to the C-Score ensure that the length of the target solution is at least some constant $k$, where $k$ is proportional to the complexity of the equation being solved. Additionally, researchers can conduct a user study that evaluates when students start

to consider an equation's solution to be too short. Then, the different terms in the C-Score can be weighted in a manner consistent with the results of the study.

## D   Experiments

Cognitive Tutor is a mathematical education software developed by Carnegie Learning based on the ACT-R theory of cognition [10]. For the evaluation of the ConPoLe model on the equations domain, the authors constructed a dataset of 290 equations (each of a different equation type) that was built from Cognitive Tutor templates. For our evaluation, a modified dataset using 284 of the Cognitive Tutor equation types was used, as 6 of the equations are unsolvable. We split the dataset into a training dataset with 198 randomly selected equation templates and a test dataset with 86 held-out problems. Our synthesis system was run for 25 iterations with $10^5$ recognition model training steps and an enumeration timeout of 1000s on 95 CPUs. Evaluation on the held-out test dataset was conducted once every 3 iterations.

As described in their respective papers, ConPoLe and Lemma were originally trained with (infix-format) equations sampled from these Cognitive Tutor equation templates that had their constants chosen at random. Both ConPoLe and Lemma were trained for $10^7$ environment steps, which are defined as queries where the environment indicates whether a problem has been solved or lists all allowed actions in the next step [7]. This training procedure results in $\approx 10^6$ equations generated from the 290 Cognitive Tutor templates by replacing equation constants. DreamCoder uses only the templates as solving one equation via a program in the DSL solves all equations in that template. As per the authors of ConPoLe and Lemma, their models were exposed to these equation templates in training, although since their training data was generated by random replacements of each constant in the template, it is highly unlikely these exact equations appeared in their training data. Hence, ConPoLe and Lemma were not trained on this specific training set, although the problems from both our training and the testing set are from the same equation template distribution that ConPoLe and Lemma were trained on.

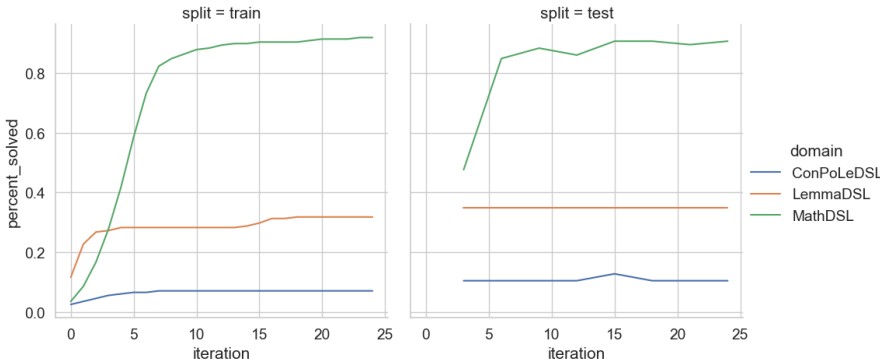

Figure 3: Percentage of tasks solved across 25 iterations for DreamCoder experiments using different DSLs.

For each model, the solutions used for the C-scores were generated as follows: for ConPoLe and Lemma, each step discovered by the model was selected as a solution step, and its C-score was computed accordingly. DreamCoder's solutions were programs in the MathDSL, ConPoLeDSL, and LemmaDSL that comprised many subprograms utilizing its learned abstractions (equation-solving strategies). These abstractions often solve equation templates from beginning to end in a single program. We make the first step in our solution the initial equation passed to the program. The abstractions are composed of several lambda functions generated by Stitch via lambda-aware unification, comprising a chain of function calls to primitive functions in the DSL. While generating solutions from programs for measuring conciseness, if the result of the lambda function is not passed as an argument to a DSL primitive function (which further simplifies the expression), we save the result of a lambda function call as a step in our final solution. By doing this, we ensure that only the highest-level lambda functions are considered while generating solution steps, and not intermediate results later simplified by other primitives. Hence, our solutions utilizing these abstractions have

multiple steps, even when the abstractions by themselves can directly solve the equation template. Examples of discovered abstractions are listed in Appendix E.

However, when parsing DreamCoder solutions by evaluating the outputs of intermediate lambda functions in the solution program, we observe that occasionally, an abstraction may be applied to the equation that does not modify the equation state at all. In such cases, DreamCoder produces duplicate steps in the solution since the program "halts" at a certain equation state before modifying it with other abstractions later in the program. If we perform a post-processing step that de-duplicates the steps in the equation solution, the conciseness metric is observed to be as follows in Table 4.

| Model Name | Training Set Accuracy | Testing Set Accuracy | Training Set Mean C-Score | Testing Set Mean C-Score |
|---|---|---|---|---|
| DreamCoder + Stitch + MathDSL | **0.9192** | **0.9070** | **0.6237** | **0.5521** |
| Lemma* | 0.7980 | 0.8488 | 0.5758 | 0.4752 |
| ConPoLe* | 0.8182 | 0.8372 | *0.0* | *0.0* |
| DreamCoder + Stitch + ConPoLeDSL | 0.0707 | 0.1047 | -0.4806 | -0.7778 |
| DreamCoder + Stitch + LemmaDSL | 0.3182 | 0.3488 | 0.5836 | 0.4874 |

Table 4: Accuracy and C-Scores (out of 198 randomly sampled training set problems and 86 test set problems) after performing a post-processing de-duplication step on DreamCoder solutions.

Mean C-Scores were computed for each model as follows: for each data set, the intersection of the set of programs solved by the baseline model (ConPoLe) and the target model (Lemma, or DreamCoder + Stitch + {MathDSL/ConPoLeDSL/LemmaDSL}), $\mathcal{S}$, was computed, and the C-Scores computed for each pair of solutions to the problems in $\mathcal{S}$. Then, the mean C-Score was computed by taking the average of C-Scores of tasks in $\mathcal{S}$.

# E  DreamCoder Abstractions for Equation Solving

This section describes some of the useful abstractions or equation-solving strategies that are generated by Stitch and learned by DreamCoder in different domain-specific languages at the end of 25 iterations. The first argument passed to the primitive is a string containing an equation in prefix form, and the second argument is an integer containing the index of the subtree on which the function acts.

Note that whenever the term is enclosed by square brackets (for example, $[x = A + B]$), this means that the term has had the simplify operation applied to it in its result (for example, if $A = 3$ and $B = 5$ in the original equation, executing the abstraction generates $x = 8$). The Conversion Formula Examples columns in Tables 5, 6, and 7 show an example transformation that can occur when the program is applied to a specific equation template. The output may differ if the same program is applied to a different equation template. A more detailed description of the ConPoLe [7] and Lemma [5] primitives can be found in the original works.

| Abstraction | Conversion Formula Examples | Description |
|---|---|---|
| `#(lambda (#(lambda (simplify (dist (#(lambda (swap (simplify $0 0) 0)) (rrotate (swap (div (#(lambda (simplify (dist (rrotate $0 1) 1) 0)) (mult (swap (#(lambda (simplify (dist (rrotate $0 1) 1) 0)) $0) 4) 3)) 3) 5) 4)) 1) 0)) (sub $0 5)))` | $(A/x) + B = C \rightarrow x = [A/(C - B)]$ | Subtract $B$ from both sides, simplifies the equation, then multiplies both sides by $x$, and then simplifying to calculate $x$'s value. |
| `#(lambda (#(lambda (lrotate (swap (#(lambda (swap (simplify $0 0) 0)) $0) 1) 1)) (#(lambda (simplify (dist (rrotate $0 1) 1) 0)) (add $0 5))))` | $Ax - B = C \rightarrow Ax = [C + B]$ | Add $B$ to both sides, eliminate $B$ from left-hand side and simplify right-hand side. |
| `#(lambda (#(lambda (swap (simplify (rrotate (swap (div (swap (simplify (rrotate $0 4) 0) 0) 3) 5) 4) 0) 0)) (#(lambda (simplify (dist (rrotate $0 1) 1) 0)) (sub $0 5))))` | $Ax + B = C \rightarrow x = [(C - B)/A]$ | Subtract $B$ from both sides, simplify both left-hand and right-hand sides, and then divide by $A$. |

Table 5: Examples of Abstractions in MathDSL

| Abstraction | Conversion Formula Examples | Description |
| --- | --- | --- |
| `#(lambda (#(lambda (multone (eval (eval (dist (eval (refl $0 0) 3) 1) 3) 2) 1)) (#(lambda (eval (refl $0 0) 1)) $0)))` | $Ax + [1-A]x = C + D \to x = [C+D]$ | Using distributivity, group the $x$ terms, and simplify. |
| `#(lambda (eval (refl $0 0) 1))` | $B + C = x \to x = [B+C]$ | Evaluate the expression and flip the order of equality |
| `#(lambda (multone (eval (eval (dist (eval (refl $0 0) 3) 1) 3) 2) 1))` | $(A+B)/C = x \to x = [(A+B)/C]$ | Flip the order of equality, and then evaluate repeatedly until equation is in its simplest form. |

Table 6: Examples of Abstractions in ConPoLeDSL

| Abstraction | Conversion Formula Examples | Description |
| --- | --- | --- |
| `#(lambda (multone (dist-dist-eval-eval-eval-eval -multone $0 0) 1))` | $Ax + [1-A]x = C + D \to x = [C+D]$ | Apply the distributive property, then evaluate the expression until it is in its simplest form. |
| `#(lambda (div-eval-comm-assoc-eval-multone (sub-assoc-eval-eval-add0 $0 5) 2))` | $Ax + B = C \to x = C - B.$ | Subtract $B$ from both sides, simplify the left hand side, then divide both sides by $A$ and simplify. |
| `#(lambda (#(lambda (div-eval-comm-assoc-eval -multone (assoc-eval-add0 $0 0) 2)) (#(lambda (refl (eval-eval $0 2) 0)) (dist $0 2))))` | $A = Bx - Cx \to x = A/(B-C).$ | Rearrange terms using the distributive property on the right side, then evaluate repeatedly and flip the order of equality. |

Table 7: Examples of Abstractions in LemmaDSL

