# OpenReview forum: "MathDSL: A Domain-Specific Language for Concise Mathematical Solutions Via Program Synthesis"
_NeurIPS.cc/2024/Workshop/MATH-AI — MATH-AI 24_

### Official Review · Reviewer_KHQt · 2024-10-06
**A good research idea but need more information to prove the novelty**

**Rating:** 6
**Confidence:** 5

**Review:**

Pro:

1. The paper presents an innovative and practical methodology for solving mathematical equations using a domain-specific language. The use of program synthesis and abstractions allows for the generation of concise and human-interpretable solutions.
2. The paper provides insightful empirical findings, demonstrating the superiority of MathDSL over other methods in terms of accuracy and conciseness of generated solutions. The comparison with ConPoLe and Lemma provides a strong basis for the claims made by the authors.
3. The paper is well-structured and provides a clear review of relevant literature. The introduction and background sections provide sufficient context for understanding the motivation and significance of the research.

Cons:

1. Can you provide more details on the training process of the models? How were ConPoLe and Lemma trained on the equation templates? How were the models evaluated on the test dataset?
2. Could you provide more information on the dataset used in the experiments? How were the equation templates selected? What were the characteristics of the equations in the dataset?
3. Can you discuss the potential solutions to address the limitations of the conciseness metric? How could the metric be refined to better account for pedagogical clarity and solution complexity?
4. Lack of novelty: While MathDSL is a novel approach, the paper does not clearly highlight its novelty compared to existing DSLs and program synthesis systems
5. The evaluation of MathDSL is limited to linear equations, and it would be valuable to see its performance in more complex mathematical domains.
6. The paper only compares MathDSL with ConPoLeDSL and LemmaDSL, and it would be beneficial to compare its performance with other existing DSLs as well.

---

### Official Review · Reviewer_xenG · 2024-10-08
**A novel DSL that makes math equation-solving more concise while using way fewer equation templates**

**Rating:** 7
**Confidence:** 4

**Review:**

**Summary:**

This paper presents a new domain-specific language, MathDSL, that when used with DreamCoder, results in highly-accurate and concise solutions to equations. The authors include an analysis of various set ups comparing the efficacy of DreamCoder with three different DSLs. The paper also presents a new AST-based metric measuring the conciseness of solutions, enabling us to measure the efficacy of DSLs beyond their plain accuracies.

**Strengths**
* The paper presents a new DSL that can be used with various systems to improve equation-solving accuracy, conciseness, and interpretability.
* The authors present a new AST-based metric that lets us measure the efficacy of an equation-solving system by measuring conciseness, i.e. how short a solution is with respect to the size of the equation trees.
* The Appendix presents an in-depth discussion of the weaknesses of the proposed evaluation metric, enabling future researchers to know of gaps they can focus on addressing.
* The Appendix includes a lot of detail about MathDSL, the program synthesis system, their experiments, and their conciseness metric. This makes the results from the paper reproducible, and makes it easier for other researchers to build upon the work of MathDSL.

**Weaknesses**
* While the authors discussing the weaknesses of the conciseness metric is notable, the issues they highlight are important, and possibly affect the scores that the different systems they compare receive.
* Linear equation-solving is a relatively narrow area, and while MathDSL presents an interesting new language for its purpose, the system would have to be generalized a lot before being broadly impactful.

---

### Official Review · Reviewer_tKYk · 2024-10-08
**The authors introduce MathDSL, a novel domain-specific language for expressing solutions to linear equations, and demonstrates that program synthesis using MathDSL with the DreamCoder framework substantially outperforms reinforcement learning approaches in terms of accuracy, solution conciseness, and data efficiency. The paper also proposes a metric for quantifying solution conciseness.**

**Rating:** 7
**Confidence:** 4

**Review:**

This paper introduces MathDSL, a domain-specific language (DSL) designed to express linear equation solutions concisely. When used within the DreamCoder program synthesis framework, MathDSL generates more accurate and concise equation solutions than state-of-the-art reinforcement learning-based approaches like ConPoLe and Lemma. The paper also introduces a quantitative metric for measuring the conciseness of equation solutions. Experiments show that combining MathDSL with DreamCoder outperforms ConPoLe and Lemma in terms of solution accuracy and conciseness while requiring substantially less training data. The interpretable abstractions generated by DreamCoder using MathDSL could also have applications in mathematical education.

Pros
- MathDSL enables concise expression of equation solutions and strong empirical performance when used with DreamCoder.
DreamCoder + MathDSL substantially outperforms RL methods in accuracy, conciseness, and data efficiency ($\sim$200 templates vs $\sim 10^6$ equations).
- The conciseness metric enables quantitative comparisons of solution quality between methods.
- Interpretable abstractions are generated, which could be used to provide solution hints to students.

Cons
- Experiments are limited to linear equations; evaluating more complex domains would test generality.
- A user study with math students could help evaluate the interpretability and educational value of the learned abstractions.
- The conciseness metric has some limitations (e.g. favoring short solutions over more gradual ones); further refinements could be explored.

---

### Official Review · Reviewer_NNYt · 2024-10-09
**Well written paper for a narrow domain of limited usefulness**

**Rating:** 6
**Confidence:** 4

**Review:**

This paper does two things to solve problems in the ConPoLe equations benchmark on equational reasoning:
* Apply the DreamCoder learning algorithm (with the faster Stitch component)
* Change the DSL to a new MathDSL to better support reasoning

They show that the two combined perform better than the two previous methods, Lemma and ConPoLe.  They also have an ablation showing that the new MathDSL is very important, in that DreamCoder + Stitch does worse with the old DSL.

The paper is clear and tidy.  It is well written and they get their point across.

I have to admit, I'm a bit disappointed by the results of this paper.  DreamCoder has always seemed like an interesting approach to problems like this, but the fact that it is so sensitive to the initial DSL is very disappointing, especially since the point of DreamCoder (and Stitch) is that it can learn new DSLs.

So it seems that the main contribution of this paper was the new MathDSL for reinforcement learning on a toy problem that frankly is not interesting by itself.  One can easily use existing software, or even write a custom Python program, to solve problems in this toy benchmark.  Unfortunately, I don't see what we have learned from this experiment that is useful outside of this toy domain.  Does this paper hold clues for how to design DSLs for other mathematical domains?  Does this paper present ideas for how to make DSLs that jumpstart DreamCoder's performance on other problems?  It is not even clear to me that the ideas from this paper would directly help in the very similar toy benchmark INT (https://github.com/albertqjiang/INT) which already has strong solutions (https://arxiv.org/abs/2206.00702).

Another thing I've noticed is that the authors show that DreamCoder + Stitch + MathDSL does better than Lemma and CanPoLe, and they additionally ablate DreamCoder + Stitch + the DSLs for Lemma and CanPoLe.  However, what about Lemma + MathDSL or CanPoLe + MathDSL?  I wonder if getting rid of DreamCoder here, but keeping the new MathDSL would do even better.  I can't tell if DreamCoder + Stitch is any benefit here, or if the only benefit is just the new MathDSL.

So in summary, this is a well-written paper that presents a good experiment.  Unfortunately, the results are mediocre and are unlikely to impact this field at all.

---

### Official Review · Reviewer_oS9d · 2024-10-09
**Review for MathDSL**

**Rating:** 7
**Confidence:** 3

**Review:**

The paper presents MathDSL, a domain-specific language designed to solve linear equations through program synthesis. By integrating MathDSL with the DreamCoder system, the approach outperforms existing methods like ConPoLe and Lemma in terms of both accuracy and solution conciseness. MathDSL generates human-interpretable, reusable abstractions that can be applied to educational tools and automated reasoning systems. It also introduces a metric to measure the conciseness of solutions.

One advantage of MathDSL is its ability to generate concise solutions compared to other systems like ConPoLe and Lemma. By reducing unnecessary steps, it produces solutions that are clearer and easier to interpret. Another benefit is its higher accuracy in solving linear equations, achieving a testing accuracy of 90.7%, which is an improvement over previous methods. Also, MathDSL allows for the creation of human-interpretable abstractions, which might be useful for other downstream applications such as education.

However, MathDSL has some drawbacks: First, it is currently limited to solving linear equations, meaning that its use is restricted unless extended to other mathematical domains. The conciseness metric, while useful, can reward solutions that are overly compressed, potentially skipping important steps that aid understanding, especially in an educational setting. Also, developing a domain-specific language like MathDSL involves significant work, such as defining primitives and abstractions, which may be difficult for more complex math problems.

---

### Decision · Program_Chairs · 2024-10-09

Accept